# *Myoparr*-Associated and -Independent Multiple Roles of Heterogeneous Nuclear Ribonucleoprotein K during Skeletal Muscle Cell Differentiation

**DOI:** 10.3390/ijms23010108

**Published:** 2021-12-22

**Authors:** Keisuke Hitachi, Yuri Kiyofuji, Masashi Nakatani, Kunihiro Tsuchida

**Affiliations:** 1Division for Therapies against Intractable Diseases, Institute for Comprehensive Medical Science (ICMS), Fujita Health University, Toyoake 470-1192, Japan; hkeisuke@fujita-hu.ac.jp (K.H.); nanbyou@fujita-hu.ac.jp (Y.K.); 2Faculty of Rehabilitation and Care, Seijoh University, Tokai 476-8588, Japan; nakatani@seijoh-u.ac.jp

**Keywords:** transcriptional regulation, myogenic differentiation, RNA-binding protein, endoplasmic reticulum stress

## Abstract

RNA-binding proteins (RBPs) regulate cell physiology via the formation of ribonucleic-protein complexes with coding and non-coding RNAs. RBPs have multiple functions in the same cells; however, the precise mechanism through which their pleiotropic functions are determined remains unknown. In this study, we revealed the multiple inhibitory functions of heterogeneous nuclear ribonucleoprotein K (hnRNPK) for myogenic differentiation. We first identified hnRNPK as a lncRNA *Myoparr* binding protein. Gain- and loss-of-function experiments showed that hnRNPK repressed the expression of *myogenin* at the transcriptional level. The hnRNPK-binding region of *Myoparr* was required to repress *myogenin* expression. Moreover, hnRNPK repressed the expression of a set of genes coding for aminoacyl-tRNA synthetases in a *Myoparr*-independent manner. Mechanistically, hnRNPK regulated the eIF2α/Atf4 pathway, one branch of the intrinsic pathways of the endoplasmic reticulum sensors, in differentiating myoblasts. Thus, our findings demonstrate that hnRNPK plays lncRNA-associated and -independent multiple roles during myogenic differentiation, indicating that the analysis of lncRNA-binding proteins will be useful for elucidating both the physiological functions of lncRNAs and the multiple functions of RBPs.

## 1. Introduction

Long non-coding RNAs (lncRNAs), which are >200 nucleotides (nt) in length and not translated into proteins, are emerging as important regulators in diverse biological processes including transcription, splicing, RNA stability, and translation [1]. LncRNAs are pervasively transcribed from the noncoding genomic DNA, cis-regulatory regions including promoter and enhancer, introns, 3′ untranslated regions, and repetitive sequences [2]. LncRNAs are also expressed from the antisense direction of the coding genomic DNA [3]. Thus, most of the genomic regions have the potential to express lncRNAs. Thus far, more than 260,000 lncRNA genes are registered in the LncBook, which is a curated knowledge-based database for human lncRNAs [4]. Since lncRNAs exert their molecular functions by interacting with proteins, mRNAs, or microRNAs [5], their molecular functions differ widely, depending on the interacting partners.

As lncRNA-interacting factors, RNA-binding proteins (RBPs) are essential to determine the molecular function of each lncRNA [1]. In the human genome, more than 1500 genes encode RBPs [6]. RBPs consist of ribonucleoprotein complexes together with lncRNAs to regulate various biological aspects. For example, the association of Ddx5/Ddx17 with lncRNAs such as *SRA*, *mrhl*, *MeXis*, or *Myoparr* is required to activate the expression of downstream genes [7,8,9,10]. RBPs including NONO, SFPQ, FUS, and RBM14 associate with *Neat1*, which is a highly abundant lncRNA in mammals, to form a large membrane-less structure paraspeckle in the nucleus [11]. A ubiquitously expressed RBP, known as human antigen R (HuR), associates with lncRNAs and regulates their stability; HuR increases the cytosolic *linc-MD1* levels in skeletal muscle cells [12], whereas it promotes the decay of *lincRNA-p21* in HeLa cells [13]. Additionally, RBPs are also involved in RNA splicing, polyadenylation, RNA transport, and translation [14]. The majority of RBPs are involved in multiple biological processes in concert with lncRNAs [15,16,17], and mutations in genes coding for RBPs are associated with human genetic disorders [18].

Heterogeneous nuclear ribonucleoprotein K (hnRNPK), a member of the heterogeneous nuclear ribonucleoprotein family, has multiple roles including chromatin remodeling, transcription, RNA splicing, and translation [19,20]. hnRNPK acts together with *lincRNA-p21*, *EWSAT1*, and *lncRNA-OG* to regulate the expression of downstream genes in mouse embryonic fibroblasts, Ewing sarcoma, and bone marrow-derived mesenchymal stem cells [21,22,23]. However, the molecular function of hnRNPK in skeletal muscle cells has not been fully elucidated. LncRNA *Myoparr* is an essential regulator of skeletal muscle cell proliferation and differentiation [8]. *Myoparr* shares the same promoter region with the *myogenin* gene and activates *myogenin* expression by promoting the interaction between Ddx17 and histone acetyltransferase PCAF [8]. We previously identified hnRNPK as a candidate for *Myoparr*-associated protein in skeletal muscle cells [8], suggesting the unique functions of *Myoparr*-associated hnRNPK during myogenic differentiation. In the present study, we revealed the inhibitory role of hnRNPK as a *Myoparr*-associated protein in skeletal muscle cell differentiation. Moreover, by comparing the downstream genes regulated by *Myoparr* and hnRNPK, we also found a *Myoparr*-independent role of hnRNPK during myogenic differentiation. Our findings reveal that hnRNPK plays *Myoparr*-associated and -independent multiple roles in skeletal muscle cell differentiation and will contribute to elucidating the complex roles of RBPs in cell differentiation.

## 2. Results

### 2.1. Identification of hnRNPK as a Myoparr-Binding Protein in Skeletal Muscle Cells

Our proteomics analysis identified both hnRNPK and TIAR as candidates for *Myoparr*-associated proteins [8]. To reveal whether hnRNPK and TIAR are associated with the *Myoparr* function during myogenic differentiation, we first examined the specific interaction of *Myoparr* with endogenous hnRNPK and TIAR. In vitro synthesized *Myoparr* was labeled with 5-bromouridine (BrU) and mixed with the nuclear extract from differentiating C2C12 myoblasts. After purification of *Myoparr* by immunoprecipitation with a BrdU antibody, specific binding between the *Myoparr* and hnRNPK protein was confirmed by immunoblotting (Figure 1A). Since binding between *EGFP* mRNA and hnRNPK was not observed, *EGFP* mRNA could work as a negative control. Intracellular binding between endogenous *Myoparr* and hnRNPK was shown by RNA immunoprecipitation using an hnRNPK-specific antibody without crosslinking (Figure 1B,C). This enrichment of *Myoparr* by hnRNPK was stronger than that of *Xist* or *Neat1*, both of which interact with hnRNPK in other cells [24,25]. Although the TIAR protein was retrieved by synthesized *Myoparr*, intracellular endogenous interaction between *Myoparr* and TIAR was not observed (Appendix A). These results suggest that hnRNPK has a function associated with *Myoparr*-binding during skeletal muscle differentiation.

### 2.2. hnRNPK Represses the Expression of Myogenin in Differentiating Myoblasts

During myogenic differentiation, the expression of *Myoparr* gradually increases and is required to activate the expression of *myogenin* [8]. Thus, we examined the changes in the expression of hnRNPK during C2C12 cell differentiation. Although the expression of myogenin was highly increased after myogenic induction, the expression of hnRNPK gradually decreased (Figure 2A). During myogenesis in mouse embryos, the expression of *hnRNPK* decreased when the expression levels of *myogenin* reached too high (Figure 2B,C), demonstrating the correlation of the expression for those two genes. We next knocked down *hnRNPK* using the small interfering RNAs (siRNAs) in differentiating C2C12 cells. An immunocytochemistry analysis of the myogenin expression after *hnRNPK* knockdown (KD) by two distinct siRNAs did not show a sufficient increase in the ratio of myogenin-positive cells compared to the control (Figure 2D). However, we observed that the number of cells with a high intensity of myogenin signal was increased by *hnRNPK* KD (Figure 2D,E). In addition, western blotting analyses showed that *hnRNPK* KD was associated with a significant increase in the expression of myogenin (Figure 2F–H). We also confirmed that *hnRNPK* KD increased the expression of myogenin in differentiating mouse primary myoblasts (Figure 2I). These results indicate that although *hnRNPK* KD is not enough to induce the expression of myogenin in cells where the expression of myogenin is not intrinsically present, *hnRNPK* KD increases the expression of myogenin in cells with intrinsic myogenin expression.

We observed that the expression of *myogenin* was significantly increased by *hnRNPK* KD (Figure 2J,K). Although the differences were not statistically significant, the *Myoparr* expression levels also tended to be increased by *hnRNPK* KD (Figure 2L). From these results, we surmise that hnRNPK negatively regulates the expression of *myogenin* at the transcriptional level. The effect of hnRNPK on the transcription of *myogenin* was examined using the *myogenin*-promoter-driven luciferase assay. The overexpression of hnRNPK in differentiating C2C12 cells decreased the *myogenin* promoter activity (Figure 2M), indicating that hnRNPK negatively regulates the expression of *myogenin* via the *myogenin* promoter.

### 2.3. The hnRNPK-Binding Region of Myoparr Is Required to Repress Myogenin Expression

The hnRNPK-binding region of *Myoparr* was determined by RNA pull-down experiments using the various forms of *Myoparr* (Figure 3A). Figure 3B showed that hnRNPK bound to the full-length sense-strand of *Myoparr* (#3 in Figure 3B), but not to the full-length antisense-strand of *Myoparr* (#4). Although deletion of the 5′-region (#5 and 6) or 3′-region (#2 and 7) of *Myoparr* did not affect binding to hnRNPK, the deletion of the 3′-half of *Myoparr* (#1) completely diminished the binding to hnRNPK (Figure 3B). These results indicate that an approximately 300-nt region (613–952 nt) of *Myoparr* is indispensable for binding to hnRNPK. Searching the motif sequence of RBPs from the 300-nt region revealed that there are eight ccawmcc motifs, which are recognized by hnRNPK (Figure 3C). The deletion of the motifs (660–729 nt) from the full-length of *Myoparr* markedly weakened the binding to hnRNPK (#8 in Figure 3B). Thus, the ccawmcc motifs on *Myoparr* were shown to be required for binding to hnRNPK.

To clarify whether *Myoparr* is involved in regulating the expression of *myogenin* by hnRNPK, we examined the effect of the ccawmcc motif on the *myogenin* promoter activity. To imitate a chromatin structure and epigenetic regulation on the plasmid DNA [27], the upstream region of *myogenin* (−1649 to +52) including *Myoparr* was cloned into an episomal luciferase vector. In accordance with our previous findings [8], the *myogenin* promoter showed high activity in the presence of *Myoparr* (−1650-Luc) in comparison to the −242-Luc construct, which only contains the *myogenin* promoter region in differentiating myoblasts (Figure 3D). Intriguingly, the activity of the −1650-Luc construct was further enhanced by the deletion of a region of approximately 70 bp (−1650∆ccawmcc-Luc), which corresponds to the 660–729 nt region on *Myoparr* (Figure 3D). These results indicate that the hnRNPK-binding region of *Myoparr* is required to repress the expression of *myogenin* during skeletal muscle differentiation.

### 2.4. hnRNPK Inhibits Skeletal Muscle Differentiation but Is Required for Normal Myotube Formation

The inhibitory role of hnRNPK in the expression of *myogenin* possibly via the ccawmcc motif on *Myoparr* suggested that hnRNPK and *Myoparr* have common downstream genes. Our RNA-Seq analysis [28] revealed that *hnRNPK* KD significantly increased the expression of 226 genes and significantly decreased the expression of 190 genes. We compared the downstream genes regulated by *hnRNPK* KD and *Myoparr* KD, and the comparative heatmap analysis showed that the genes regulated by *hnRNPK* KD showed the opposite direction to the *Myoparr* KD (Figure 4A). Twenty percent of genes (84/416) altered by *hnRNPK* KD overlapped with genes regulated by the *Myoparr* KD (Figure 4B and Appendix A). The intersection of these genes was 12.3-fold greater than that expected by chance (*p* = 1.393037 × 10^−45^). Although these 84 genes showed a low correlation coefficient (R = 0.0758323), we observed a negative correlation trend for one segment including *myogenin*; the expression of 50 genes belonging to this segment was increased by *hnRNPK* KD and decreased by *Myoparr* KD (red frame in Figure 4C). These genes were enriched in sarcomere organization, myofibril assembly, and muscle contraction categories in GO terms (Figure 4D), indicating that hnRNPK inhibits myogenic differentiation and maturation.

In accordance with the results of the RNA-Seq analysis, we observed a significant increase in the expression of Myod, one of the master regulators of myogenesis, and Myosin heavy chain (MHC), which is a later marker of myogenic differentiation and maturation, in the early stages of differentiation with *hnRNPK* KD (Figure 4E), indicating that the *hnRNPK* KD causes premature differentiation of myoblasts. Although *hnRNPK* KD increased MHC expression in the late stages of differentiation (Figure 4F), *hnRNPK* KD did not affect the percentage of differentiated cells; this was shown by the fusion index (Figure 4G) as well as the results from immunocytochemistry to detect myogenin (Figure 2D). Instead, we observed the appearance of locally spherical myotubes following *hnRNPK* KD. These differed from the normal tube-shaped myotubes (Figure 4G). Thus, these results indicate that hnRNPK is required for normal myotube formation, possibly through the inhibitory effect on the premature differentiation of myoblasts at early stages of differentiation.

### 2.5. hnRNPK Represses the Expression of Aminoacyl-tRNA Synthetases via the eIF2α/Atf4 Pathway

The Venn diagram in Figure 4B indicates that 332 genes regulated by *hnRNPK* KD were *Myoparr*-independent, suggesting the *Myoparr*-independent role of hnRNPK in differentiating myoblasts. We performed an enrichment analysis of genes regulated by *hnRNPK* KD and compared them with genes regulated by *Myoparr* KD. As we have reported [8], genes related to cell cycle and cell division were only enriched in genes regulated by *Myoparr* KD (Figure 5A). Skeletal muscle-associated genes were regulated by both *Myoparr* KD and *hnRNPK* KD (Figure 5A). Intriguingly, genes coding for aminoacyl-tRNA synthetases were regulated specifically by *hnRNPK* KD (red frame in Figure 5A). In mice, there are two-types of aminoacyl-tRNA synthetases: cytosolic and mitochondrial aminoacyl-tRNA synthetase. Our RNA-Seq analysis showed that *hnRNPK* KD significantly increased the expression of 10 genes coding for cytosolic aminoacyl-tRNA synthetases, whereas it had little effect on the expression of genes coding for mitochondrial aminoacyl-tRNA synthetases (Appendix A). To confirm the RNA-Seq results, we picked up five genes coding for cytosolic aminoacyl-tRNA synthetases, and their expression changes by *hnRNPK* KD were verified by qRT-PCR. *hnRNPK* KD, using two distinct siRNAs, significantly increased the expression of *Aars*, *Gars*, *Iars*, *Nars*, and *Sars* (Figure 5B–F). This regulation was not observed following *Myoparr* KD (Figure 5B–F), indicating that hnRNPK regulates the expression of these genes in a *Myoparr*-independent manner.

The expression of genes coding for almost all cytosolic aminoacyl-tRNA synthetases is activated by transcription factor Atf4 [29,30]. Thus, we examined whether *hnRNPK* KD altered the expression of Atf4 in skeletal muscle cells. Although not statistically significant, the *Atf4* expression in differentiating myoblasts tended to be increased by *hnRNPK* KD (Figure 5G). *Myoparr* KD did not affect the expression of *Atf4* (Figure 5G). The effect of *hnRNPK* KD was more pronounced in the expression of Atf4 protein. The amount of Atf4 protein was highly increased by *hnRNPK* KD in differentiating C2C12 cells and mouse primary myoblast (Figure 5H,I). We further examined the expression changes of other Atf4 target genes by *hnRNPK* KD in differentiating myoblasts. The expression levels of *Asns* and *Psat1*, which encode proteins related to amino acid synthesis, were significantly increased by *hnRNPK* KD (Appendix A). In addition, the expression levels of *Chop*, *Chac1*, and *Trb3*, pro-apoptosis genes, and *Gadd34*, an another Atf4 target gene, also tended to be increased by *hnRNPK* KD (Appendix A). These results suggest that hnRNPK regulates the expression of genes associated with amino acid synthesis via the expression of Atf4.

Under the condition of endoplasmic reticulum (ER) stress, *Atf4* mRNA is translated more efficiently and contributes to the restoration of cell homeostasis via the regulation of cytosolic aminoacyl-tRNA synthetases [31]. Thus, to reveal the molecular mechanism by which hnRNPK regulates the expression of Atf4, we finally focused on ER stress. Integrated stress response inhibitor (ISRIB) is an inhibitor of eIF2α, which is a downstream component of PERK signaling, one branch of the ER stress sensors. We investigated whether ISRIB treatment could suppress the increase in the expression of Atf4 induced by *hnRNPK* KD and found that ISRIB treatment completely rescued this *hnRNPK*-KD-induced increase (Figure 6A). Furthermore, ISRIB treatment abrogated the increased expression of Atf4 target genes, *Aars*, *Gars*, *Iars*, *Nars*, and *Sars* by *hnRNPK* KD (Figure 6B–F and Appendix A). Thus, these results indicate that *Myoparr*-independent hnRNPK function is the regulation of the eIF2α/Atf4 pathway during myogenic differentiation.

## 3. Discussion

RBPs have multiple molecular functions including RNA splicing, transcription, translation, RNA stability, and the formation of the nuclear structure to regulate cell proliferation, differentiation, development, and diseases [14]. Although many RBPs have multiple functions in the cells [15,16,17], it is still unclear how their pleiotropic functions are determined. In this study, we revealed novel multiple functions of hnRNPK, a member of the hnRNP family of RBPs, in skeletal muscle cells. By focusing on a lncRNA *Myoparr*-associated protein, we found that hnRNPK repressed the expression of *myogenin*, coding for one of the master regulators of muscle differentiation. Deletion of the hnRNPK-binding region of *Myoparr* activated the expression of *myogenin*. Moreover, our comparative analysis of the downstream genes of hnRNPK and *Myoparr* showed that the function of hnRNPK was pleiotropic. During myogenic differentiation, hnRNPK repressed the expression of a set of genes coding for cytosolic aminoacyl-tRNA synthetases via the eIF2α/Atf4 pathway. Taken together, our study revealed multiple inhibitory roles of hnRNPK in skeletal muscle cells: one was *Myoparr*-associated and the other was *Myoparr*-independent (Figure 7). Recently, Xu et al. reported that the deficiency of 36 amino acids in hnRNPK diminished C2C12 differentiation [32]. However, our results provided strong evidence to support that hnRNPK has an inhibitory effect on muscle differentiation. In addition, we observed the appearance of locally spherical myotubes following *hnRNPK* KD. Considering the facts that dysregulated Myod expression leads to premature myogenic differentiation [33] and results in the formation of dysfunctional myofibers in mice [34], uncoordinated increases in Myod, myogenin, and MHC expression by *hnRNPK* KD may lead to the abnormal shape of myotubes. In addition, the morphology of these myotubes closely resembles myotubes with myofibril-assembly defects [35], suggesting that hnRNPK may also be involved in the regulation of the myofibril assembly in myotubes through intrinsic fine-tuning of muscle cell differentiation both in *Myoparr*-associated and *Myoparr*-independent manner. Therefore, despite its lncRNA-associated and -independent roles in the inhibition of myogenic differentiation, hnRNPK is apparently required for the formation of normal myotubes.

We observed that *hnRNPK* KD increased myogenin protein levels more robustly than *myogenin* mRNA. The peak expression of myogenin protein was detected 1–2 days after that of *myogenin* mRNA in both in vitro and in vivo myogenesis [36], suggesting that a slight increase in *myogenin* mRNA by *hnRNPK* KD in the early stages of myogenic differentiation eventually led to a marked increase in myogenin protein. Therefore, the fine-tuning of the *myogenin* expression by hnRNPK at the early stages of differentiation may have a significant impact on the overall myogenic differentiation processes through the RNA–protein network. Intriguingly, despite the percentage of myogenin-positive cells that was not changed, the expression of myogenin protein was increased by *hnRNPK* KD. We observed that *hnRNPK* KD only increased the number of cells with a high intensity of myogenin signal. These results suggest that hnRNPK can specifically repress the expression of *myogenin* in a subset of responding cells, rather than by simply turning off the expression of *myogenin* in every myoblast. Our experiments showed that hnRNPK repressed the expression of *myogenin* at the transcriptional level, possibly via binding to the ccawmcc motif on *Myoparr*, suggesting that the existence of *Myoparr* would be necessary for hnRNPK to inhibit the expression of *myogenin*. Since *myogenin* and *Myoparr* share the same promoter region [8], *myogenin* and *Myoparr* are likely expressed in the same cells. Thus, the expression of *myogenin* would not be activated in the cells without the cell-intrinsic expression of *Myoparr*, even if *hnRNPK* is depleted in every myoblast. Further studies are required to investigate the more precise molecular mechanism by which hnRNPK regulates the expression of *myogenin* via binding to *Myoparr*.

ER stress is induced by several perturbations disrupting cell homeostasis including protein misfolding, viral infection, and changes in intracellular calcium concentration [37]. The cells recognize those stresses with three branches of ER transmembrane sensors signaling, PERK, inositol-requiring protein 1 (IRE1), and ATF6 [31]. During myogenic differentiation, ATF6 signaling was activated and led to apoptosis in myoblasts [38]. The increased phosphorylation of eIF2α, a component of PERK signaling, was observed in myoblasts at the early stage after the induction of differentiation [39]. In addition, treatment with ER stress inducers enhanced apoptosis in myoblasts, but led to efficient myogenic differentiation in the remaining surviving cells [40]. Recently, the deletion of PERK in satellite cells, which are adult muscle stem cells, was shown to inhibit myogenic differentiation and led to impaired skeletal muscle regeneration in adult mice [41], indicating that ER stress promotes myogenic differentiation. In this study, we showed that the *hnRNPK* KD in differentiating myoblasts increased the expression of Atf4 and this effect was diminished by treatment with ISRIB, a specific inhibitor of eIF2α. The expression levels of a set of genes coding for cytosolic aminoacyl-tRNA synthetases, which are regulated by Atf4, were also increased by the *hnRNPK* KD and were completely rescued after ISRIB treatment. The inhibitory effects of hnRNPK on Atf4 and cytosolic aminoacyl-tRNA synthetases were independent of *Myoparr*. Therefore, our findings suggest that hnRNPK fine-tunes the myogenic differentiation process by modulating ER stress via eIF2α/Atf4 signaling in a lncRNA-independent manner. Since hnRNPK is involved in the translational efficiency [42,43], a decrease in hnRNPK may load on the translational machinery and activate ER stress via eIF2α/Atf4 signaling in myoblasts. Alternatively, there are three other eIF2α kinases (Hri, Pkr, and Gcn2) in mammals besides PERK, thus it cannot be excluded that hnRNPK regulates eIF2α/Atf4 signaling through these upstream factors of eIF2α.

In conclusion, hnRNPK plays multiple lncRNA-dependent and -independent roles in the inhibition of myogenic differentiation. Thus, the analysis of RBPs bound to lncRNAs will be useful for elucidating both the physiological functions of lncRNAs and the complex functions of RBPs in cell differentiation. Induced ER stress including increased PERK signaling was observed in skeletal muscle biopsy samples from myotonic dystrophy 1 patients and in mdx mice, a model of Duchenne muscular dystrophy [44,45]. Moreover, mutations in genes coding for aminoacyl-tRNA synthetases are implicated in human neuromuscular disorders [46], and autoantibodies against aminoacyl-tRNA synthetases are found in autoimmune disease [47]. Collectively, downstream genes of hnRNPK are strongly associated with neuromuscular and other disorders in humans, suggesting that targeting hnRNPK to regulate the expression of these genes and signaling may become a new therapeutic strategy for human diseases.

## 4. Materials and Methods

### 4.1. Cell Cultures, siRNA Transfection, and ISRIB Treatment

A mouse myoblast cell line, C2C12, was cultured in Dulbecco’s modified Eagle medium (DMEM) supplemented with 10% fetal bovine serum at 37 °C under 5% CO_2_. Mouse primary myoblasts were isolated from lower extremity muscles of 8-week-old C57BL6J mice as described previously [48]. Myogenic differentiation of C2C12 cells and primary myoblasts were induced by replacing the medium with the differentiation medium, DMEM supplemented with 2% or 5% horse serum, respectively. Cells were transfected with 50 nM of Stealth RNAi (Thermo Fisher Scientific, Waltham, MA, USA) using Lipofectamine 3000 (Thermo Fisher Scientific, Waltham, MA, USA) or RNAiMax (Thermo Fisher Scientific, Waltham, MA, USA) according to the manufacturer’s protocol. The following siRNAs were used: Stealth RNAi siRNA negative control (negative control, Med GC, Thermo Fisher Scientific, Waltham, MA, USA), stealth RNAi for *Myoparr*, and stealth RNAi siRNAs specific for *hnRNPK* (MSS205172 and MSS205173, Thermo Fisher Scientific, Waltham, MA, USA). The siRNA sequences are listed in Appendix A. At 24 h after siRNA transfection, myogenic differentiation was induced. At 24 h or 72 h after differentiation induction, cells were collected for the analysis of RNAs and proteins. For ISRIB treatment, the differentiation medium was added either with or without 1 µM trans-ISRIB (No.16258, Cayman Chemical Company, Ann Arbor, MI, USA).

### 4.2. RNA Isolation, Reverse Transcription Reaction, and Quantitative RT-PCR

Total RNA was extracted from C2C12 cells using the ISOGEN II reagent (Nippon Gene, Tokyo, Japan) according to the manufacturer’s protocol. Total RNA from embryonic skeletal muscles were surgically isolated from the trunk region of mouse embryos at E10.5, E12.5, and E14.5. After DNase I (Thermo Fisher Scientific, Waltham, MA, USA) treatment, total RNA was used for reverse transcription reaction using SuperScript III Reverse Transcriptase (Thermo Fisher Scientific, Waltham, MA, USA) or ProtoScript II Reverse Transcriptase (New England Biolabs (NEB), Beverly, MA, USA) with random or oligo (dT) primers (Thermo Fisher Scientific, Waltham, MA, USA). Quantitative real-time PCR was conducted using SYBR Premix Ex Taq (Takara Bio Inc., Shiga, Japan) and a Thermal Cycler Dice Real Time System TP800 (Takara Bio Inc., Shiga, Japan). The results were normalized to the *Rpl26* expression, except for the analysis in mouse embryos, in which *GAPDH* expression was used for normalization. The primers used are listed in Appendix A.

### 4.3. Protein Extraction and Western Blotting

Cells were lysed in RIPA buffer (50 mM Tris-HCl (pH 8.0), 150 mM NaCl, 0.1% SDS, 1% Triton X-100, 0.5% sodium deoxycholate) containing protease inhibitors (1 mM phenylmethylsulfonyl fluoride, 1 μg/mL aprotinin, 4 μg/mL leupeptin) and phosphatase inhibitors (5 mM NaF, 5 mM β-glycerophosphate, 1 mM Na_3_VO_4_). The protein concentration was measured using a Pierce BCA Protein Assay Kit (Thermo Fischer Scientific, Waltham, MA, USA). Equal amounts of protein were used for western blotting. The following primary antibodies were used: myogenin antibody (F5D) (sc-12732, Santa Cruz Biotechnology, Dallas, TX, USA), MHC antibody (MF20, Developmental Studies Hybridoma Bank (DSHB), Iowa City, IA, USA), Myod antibody (CE-011A, Cosmo Bio Co. Ltd., Tokyo, Japan), hnRNPK antibody (#4675, Cell Signaling Technology (CST), Beverly, MA, USA), hnRNPK antibody (F45P9C7, BioLegend, San Diego, CA, USA), TIAR antibody (#8509, CST, Beverly, MA, USA), and Atf4 antibody (693901, BioLegend, San Diego, CA, USA). The following HRP-linked secondary antibodies were used: anti-mouse IgG (#7076, CST, Beverly, MA, USA), anti-rabbit IgG (#7074, CST, Beverly, MA, USA), and TrueBlot ULTRA anti-Ig HRP, Mouse (Rat) (18-8817-33, Rockland Immunochemicals Inc., Limerick, PA, USA). Can Get Signal Immunoreaction Enhancer Solution (Toyobo, Osaka, Japan) was used when necessary. The signal was detected with ImmunoStar LD reagent (FUJIFILM Wako, Osaka, Japan) using a cooled CCD camera system (Light-Capture, ATTO, Tokyo, Japan).

### 4.4. Immunofluorescence Assay

Immunofluorescence analyses of C2C12 cells were performed as previously described [8]. Briefly, 24 h or 72 h after the induction of differentiation, cells were fixed with 4% PFA and permeabilized with 0.2% Triton X-100. After blocking with 5% FBS, cells were stained with an anti-myogenin antibody (F5D, DSHB, Iowa City, IA, USA) or with an anti-MHC antibody (MF20, DSHB, Iowa City, IA, USA). The following secondary antibodies were used: Goat anti-Mouse IgG (H+L) Highly Cross-Adsorbed Secondary Antibody, Alexa Fluor 488 (Thermo Fisher Scientific, Waltham, MA, USA). Nuclei were counterstained with DAPI (Dojindo, Kumamoto, Japan). A DMI4000B microscope with a DFC350FX CCD camera (Leica, Wetzlar, Germany) was used for the visualization of signals. Images were analyzed using the ImageJ software program (ver. 1.53a).

### 4.5. Identification of Myoparr-Binding Proteins

*Myoparr*-binding proteins were collected with the RiboTrap Kit (Medical & Biological Laboratories (MBL), Aichi, Japan) using BrU labeled RNAs, as described previously [8]. BrU-labeled *Myoparr* and *EGFP* RNA were prepared using the Riboprobe System (Promega, Madison, WI, USA). Twenty-four hours after the induction of differentiation, nuclear extract was prepared from differentiating C2C12 myoblasts. The *Myoparr* or *EGFP* RNA (50 pmol) were mixed with the nuclear extract from differentiating C2C12 myoblasts for 2 h at 4 °C. The RNA–protein complexes were collected by Protein G Plus Agarose (Thermo Fisher Scientific, Waltham, MA, USA) conjugated to an anti-BrdU antibody, and proteins were eluted by adding BrdU. Purified proteins were detected by western blotting using a specific antibody, as described above.

### 4.6. RNA Immunoprecipitation and the RNA Pull-Down Assay

Immunoprecipitation of endogenous RNAs (*Myoparr*, *Xist*, or *Neat1*) was performed using the RIP-Assay Kit (MBL, Aichi, Japan) using 2 × 10^7^ C2C12 cells 48 h after the induction of differentiation, as previously described [8]. The following antibodies were used for RNA immunoprecipitation: normal rabbit IgG (#2729S, CST, Beverly, MA, USA), anti-HNRNPK pAb (RN019P, MBL, Aichi, Japan), or TIAR mAb (#8509, CST, Beverly, MA, USA). After treatment with DNase I, immunoprecipitated RNAs were used for the reverse transcription reaction. The precipitation percentage (precipitated RNA vs. input RNA) was calculated by qRT-PCR using the primers listed in Appendix A.

An RNA pull-down assay was performed with a RiboTrap Kit. Various lengths of *Myoparr* were subcloned into a pGEM-Teasy vector (Promega, Madison, WI, USA). BrU-labeled *Myoparr* (10 pmol each) was bound to Protein G Plus Agarose conjugated to an anti-BrdU antibody and mixed with the in vitro transcribed/translated hnRNPK protein. After several washing steps, the binding of hnRNPK to *Myoparr* was analyzed by western blotting using an hnRNPK antibody as described above.

### 4.7. Luciferase Reporter Assay

The upstream region of *myogenin* (−1650/+51) was PCR-amplified and cloned into the pGL4.20 vector (Promega, Madison, WI, USA). The *hnRNPK*-expressing plasmid was a kind gift from Dr. H. Okano [43]. Proliferating C2C12 cells were transfected with the *myogenin* promoter and *hnRNPK*-expressing plasmid using Lipofectamine 2000 (Thermo Fisher Scientific, Waltham, MA, USA). The total amount of DNA was kept constant by the addition of the pcDNA3 vector. Cells were collected at 24 h after the induction of differentiation and dissolved in Passive Lysis Buffer (Promega, Madison, WI, USA). The effect of hnRNPK on *myogenin* promoter was measured using a Lumat LB 9507 luminometer (Berthold Technologies, Bad Wildbad, Germany) with the Dual-Luciferase Reporter Assay System (Promega, Madison, WI, USA) according to the manufacturer’s protocol. The pGL4.74 vector (Promega, Madison, WI, USA) was used as an internal control. The relative luciferase activity is shown as the firefly to Renilla luciferase ratio.

To reconstitute the complex chromatin structure and epigenetic regulation in vitro [27], the upstream regions of *myogenin* (−242/+51 and −1650/+51) were subcloned into the episomal luciferase vector, pREP4-luc, and the −242-Luc and −1650-Luc constructs were generated. To create the −1650∆ccawmcc-Luc construct, the upstream region (−971/−902) of *myogenin* was deleted from the −1650-Luc construct. The region containing the putative hnRNPK-binding motif ccawmcc was identified by RBPmap (http://rbpmap.technion.ac.il/ 21 December 2021) [49]. The episomal luciferase vector, pREB7-Rluc, was used as an internal control for the luciferase assay. The pREP4-luc and pREP7-Rluc vectors were gifts from K. Zhao [27]. Subconfluent C2C12 cells were transfected with the indicated episomal luciferase vectors using Lipofectamine 2000. After myogenic induction, cells were collected and used to measure the relative luciferase activity.

### 4.8. Analysis of Downstream Genes Regulated by Myoparr KD and hnRNPK KD

Downstream genes regulated by *Myoparr* KD and *hnRNPK* KD were identified as described previously [28] using our RNA-Seq raw data (accession No. DRA005527). Briefly, statistical analysis of differentially expressed genes by *Myoparr* KD and *hnRNPK* KD was performed using DESeq2 ver. 1.12.4 software [50] with a Wald test (cut-offs: false discovery rate (adjusted *p*-value, padj) <0.05 and log 2 fold change >0.75 or <−0.75). A pathway analysis of significantly upregulated genes coding for cytosolic aminoacyl-tRNA synthetases was performed based on the KEGG [51]. A Gene Ontology (GO) analysis of differentially expressed genes was performed with DAVID ver. 6.8 (https://david.ncifcrf.gov/ 21 December 2021). An enrichment analysis was performed using Metascape [52].

### 4.9. Statistical Analysis

Error bars represent standard deviation. Statistical analyses were performed using unpaired two-tailed Student’s *t*-tests. For comparisons of more than two groups, a one-way ANOVA followed by Tukey’s post hoc test was performed using Prism 9 (GraphPad Software, San Diego, CA, USA). Statistical significance is reported in the figures and figure legends. *p* values of < 0.05 were considered statistically significant.

## Figures and Tables

**Figure 1 ijms-23-00108-f001:**
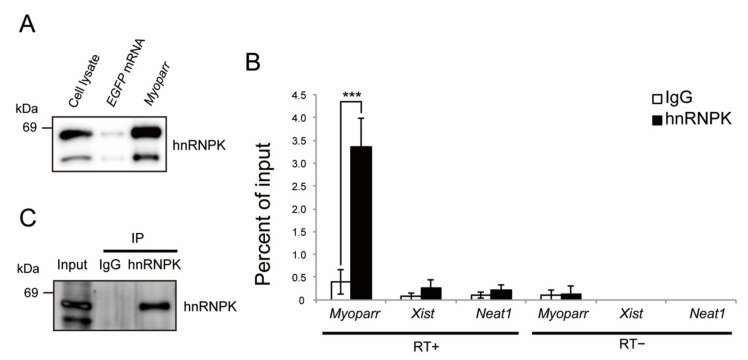
Intracellular interaction between *Myoparr* and hnRNPK in differentiating C2C12 cells. (**A**) Synthesized *Myoparr* was mixed with the C2C12 nuclear extract. Following purification of *Myoparr* by immunoprecipitation, the interaction between *Myoparr* and hnRNPK was confirmed by immunoblotting using an hnRNPK-specific antibody. *EGFP* mRNA was used for the control. Note that the smaller band would be a splicing isoform of hnRNPK [26]. (**B**) RNA immunoprecipitation for *Myoparr*, *Xist*, and *Neat1* in differentiating C2C12 cells by an hnRNPK antibody. The interaction between endogenous *Myoparr* and hnRNPK was detected by qRT-PCR two days after the differentiation induction. Normal rabbit IgG was used for the control. The presence or absence of reverse transcription reaction is shown by (RT+) or (RT−), respectively. n = 4, mean ± SD. *** *p* < 0.001. (**C**) hnRNPK protein from C2C12 cells purified by immunoprecipitation in (**B**) was confirmed by western blotting.

**Figure 2 ijms-23-00108-f002:**
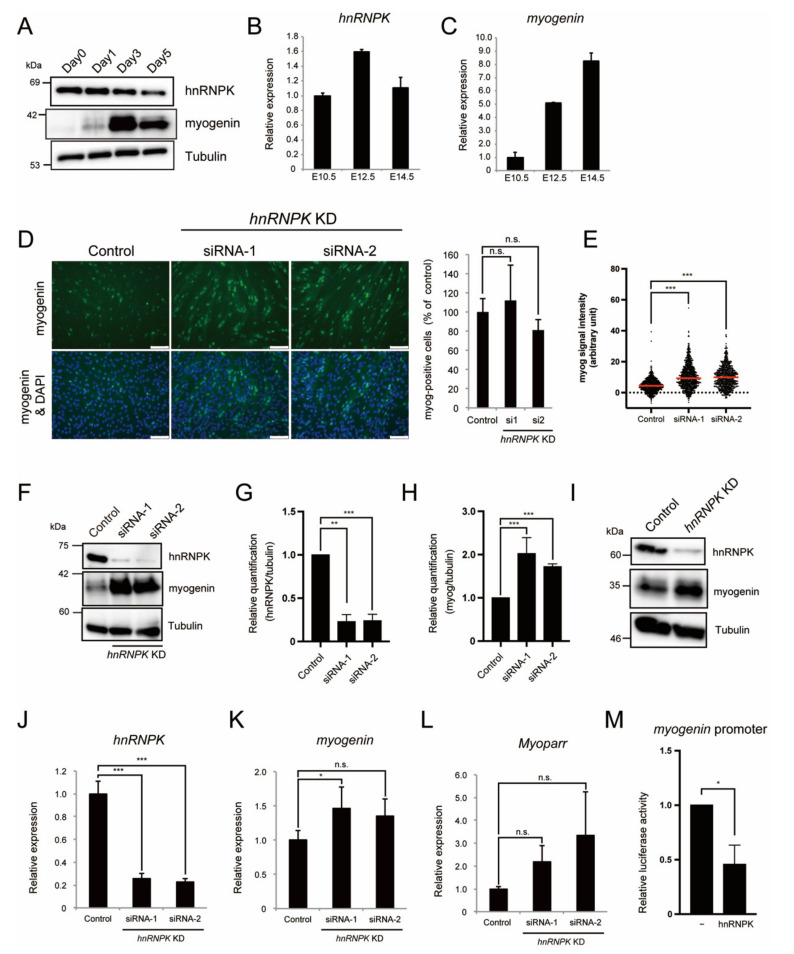
hnRNPK represses the expression of myogenin at the transcriptional level. (**A**) Western blots showing the hnRNPK and myogenin expression during C2C12 differentiation. The tubulin expression served as an internal control. (**B**,**C**) Quantitative RT–PCR for *hnRNPK* (**B**) and *myogenin* (**C**) during myogenesis in mouse embryonic skeletal muscle. The x-axis shows embryonic days. n = 3, mean ± SD. (**D**) Immunocytochemistry for myogenin 48 h after *hnRNPK* KD in C2C12 cells. Nuclei were counterstained with DAPI. Bar, 100 μm. The percentage of the myogenin-positive cells is shown as the percentage of the control. n = 4, mean ± SD. n.s., not significant. (**E**) The results of the cell count based on the signal intensity of immunocytochemistry for myogenin of (**D**). The red line indicates the median. Nine image fields were counted per sample. Control; n = 824, *hnRNPK* siRNA-1; n = 859, *hnRNPK* siRNA-2; n = 761. *** *p* < 0.001. A Mann–Whitney nonparametric test was used for comparisons between each group. (**F**) Western blots showing increased myogenin expression in differentiating C2C12 cells 48 h after *hnRNPK* KD. Blots are representative of four repeats. (**G**,**H**) Relative quantification of hnRNPK (**G**) and myogenin (**H**) from (**F**). n = 4, mean ± SD. *** *p* < 0.001, ** *p* < 0.01. (**I**) Increased myogenin expression in differentiating mouse primary myoblasts 48 h after *hnRNPK* KD. Blots are representative of three repeats. (**J**–**M**) Quantitative RT-PCR to detect the expression of *hnRNPK* (**J**), *myogenin* (**K**), and *Myoparr* (**L**) after *hnRNPK* knockdown. n = 3–4, mean ± SD. *** *p* < 0.001, * *p* < 0.05, n.s., not significant. (**M**) Exogenous hnRNPK decreased the promoter activity of *myogenin* in differentiating C2C12 cells. n = 3, mean ± SD. * *p* < 0.05.

**Figure 3 ijms-23-00108-f003:**
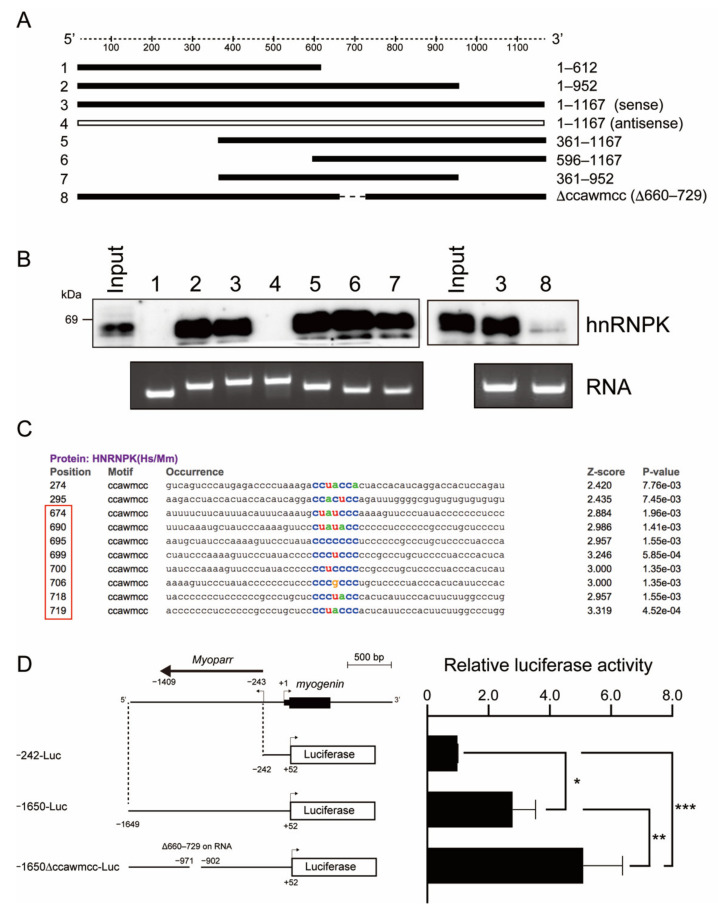
The hnRNPK region on *Myoparr* is required for repression of *myogenin* promoter activity. (**A**) A schematic diagram of various lengths of *Myoparr* used for RNA pull-down assays. (**B**) In vitro transcribed/translated hnRNPK protein was pulled down by *Myoparr* and then detected by western blotting using an hnRNPK-specific antibody. Lane numbers are consistent with (**A**). The RNA lanes indicate various forms of *Myoparr* used for RNA pull-down. (**C**) The results of an RBPmap analysis searching the motif of hnRNPK. Colored letters correspond to the hnRNPK motif. The red frame indicates the ccawmcc motifs within the region deleted in ∆ccawmcc, #8 in (**A**). (**D**) The left panel shows a schematic diagram of the constructs used for the luciferase assays. −242-Luc contains the *myogenin* promoter region. −1650-Luc contains both the *myogenin* promoter region and *Myoparr*. −1650∆ccawmcc-Luc contains the *myogenin* promoter region and *Myoparr* without the ccawmcc motifs, corresponding to #8 in A. The right panel shows the relative luciferase activities of indicated constructs in differentiating C2C12 cells. n = 4, mean ± SD. *** *p* < 0.001, ** *p* < 0.01, * *p* < 0.05.

**Figure 4 ijms-23-00108-f004:**
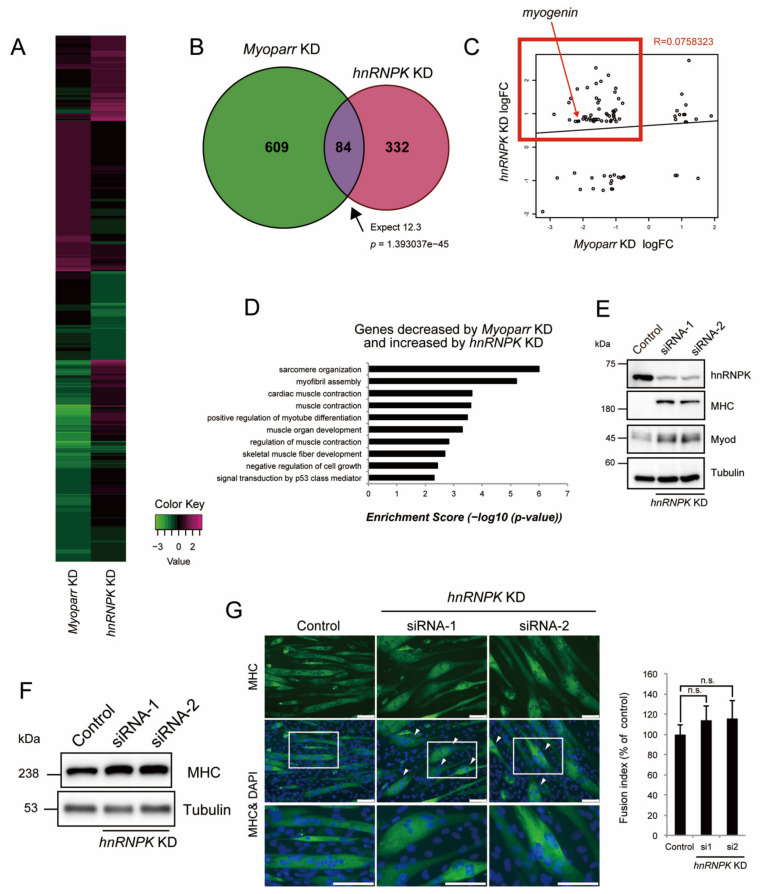
hnRNPK inhibits myogenic differentiation but is required for normal myotube formation. (**A**) A heatmap diagram showing the increased or decreased expression of genes regulated by *Myoparr* KD and *hnRNPK* KD. (**B**) The intersection of genes regulated by *Myoparr* KD and *hnRNPK* KD shows a significant (Fisher’s exact test) overlap, which is 12.3 times larger than would be expected by chance alone. (**C**) Eighty-four genes with expression levels that were significantly altered by both *Myoparr* KD and *hnRNPK* KD showed an uncorrelated pattern (R = 0.0758323, log 2 ratio scale). The red frame indicates the 50 genes for which the expression was decreased by *Myoparr* KD, but increased by *hnRNPK* KD. (**D**) Enrichment GO categories of the 50 genes corresponding to the red frame in (**C**). (**E**) Western blots showing increased MHC and Myod expression in C2C12 cells 48 h after *hnRNPK* KD. Blots are representative of three repeats. (**F**) Western blot showing the increased expression of MHC in C2C12 myotubes 96 h after *hnRNPK* KD. The tubulin expression served as an internal control. The blots shown are representative of three experiments. (**G**) Immunocytochemistry for MHC 96 h after *hnRNPK* KD. Nuclei were counterstained with DAPI. Arrowheads in the middle panels indicate locally spherical myotubes. The bottom panels are magnified views of the boxed regions in the middle panels. Bar, 100 μm. The fusion index is shown as the percent of the control. n = 3, mean ± SD. n.s., not significant.

**Figure 5 ijms-23-00108-f005:**
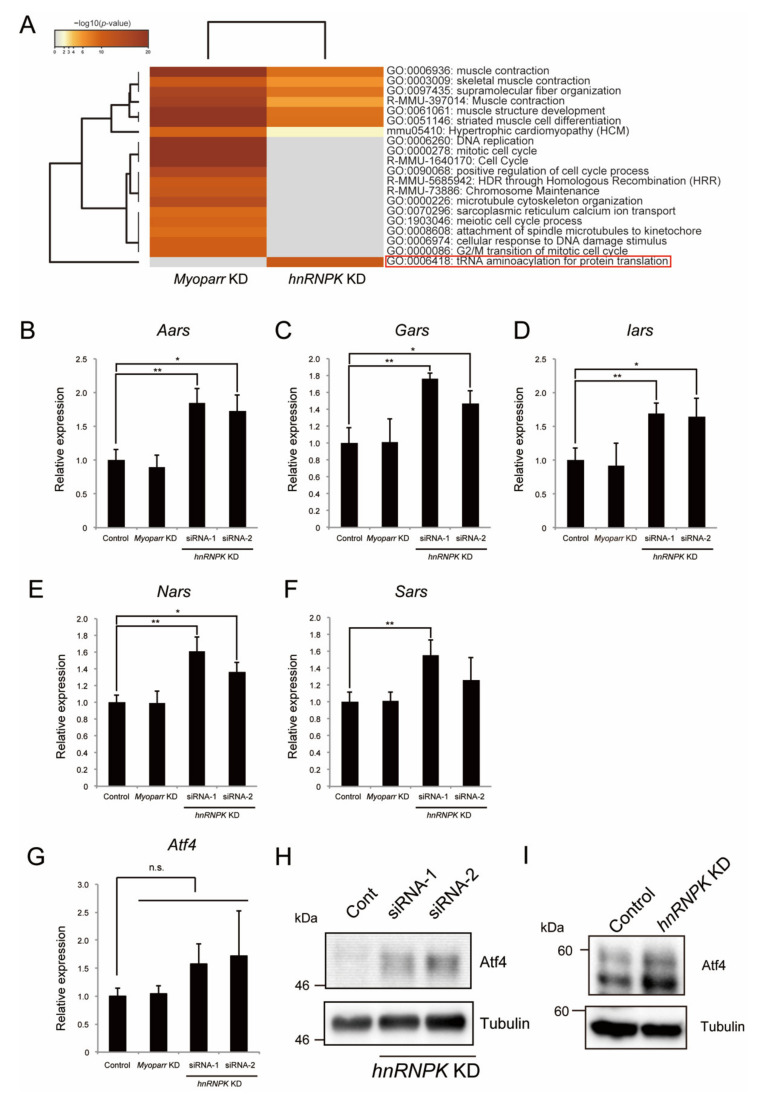
hnRNPK regulates the expression of genes coding for aminoacyl-tRNA synthetases in a *Myoparr*-independent manner. (**A**) A Metascape analysis of genes regulated by *Myoparr* KD and *hnRNPK* KD. The red frame indicates the GO terms that were specifically enriched in *hnRNPK* KD. (**B**–**G**) qRT-PCR to detect the expression of *Aars* (**B**), *Gars* (**C**), *Iars* (**D**), *Nars* (**E**), *Sars* (**F**), and *Atf4* (**G**) 48 h after either *Myoparr* KD or *hnRNPK* KD. n = 3, mean ± SD. ** *p* < 0.01, * *p* < 0.05, n.s., not significant. (**H**,**I**) Western blot showing the increased expression of Atf4 in differentiating C2C12 cells (**H**) and mouse primary myoblasts (**I**) 48 h after *hnRNPK* KD. The tubulin expression served as an internal control. Blots are representative of three experiments.

**Figure 6 ijms-23-00108-f006:**
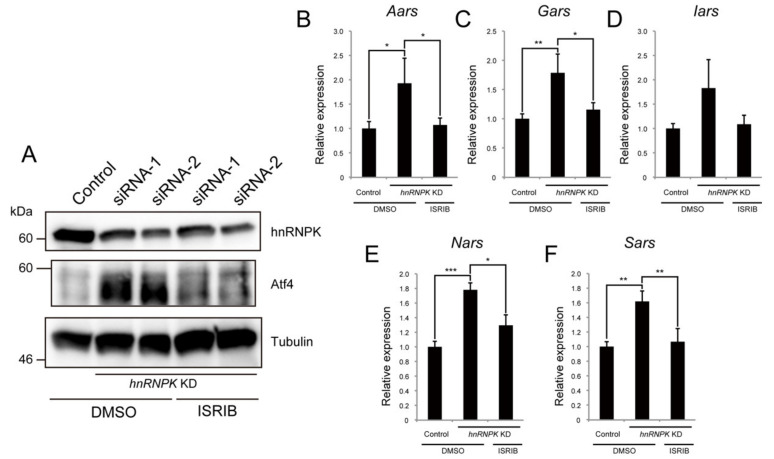
hnRNPK regulates the expression of genes coding for aminoacyl-tRNA synthetases via the eIF2α/Atf4 pathway. (**A**) Western blot showing the increased Atf4 expression 48 h after *hnRNPK* KD, and ISRIB treatment suppressed the increase. Dimethyl sulfoxide (DMSO) was used as the control. The blots shown are representative of three experiments. (**B**–**F**) qRT-PCR to detect the expression of *Aars* (**B**), *Gars* (**C**), *Iars* (**D**), *Nars* (**E**), *Sars* (**F**) 48 h after *hnRNPK* KD with or without ISRIB treatment. n = 3, mean ± SD. *** *p* < 0.001, ** *p* < 0.01, * *p* < 0.05.

**Figure 7 ijms-23-00108-f007:**
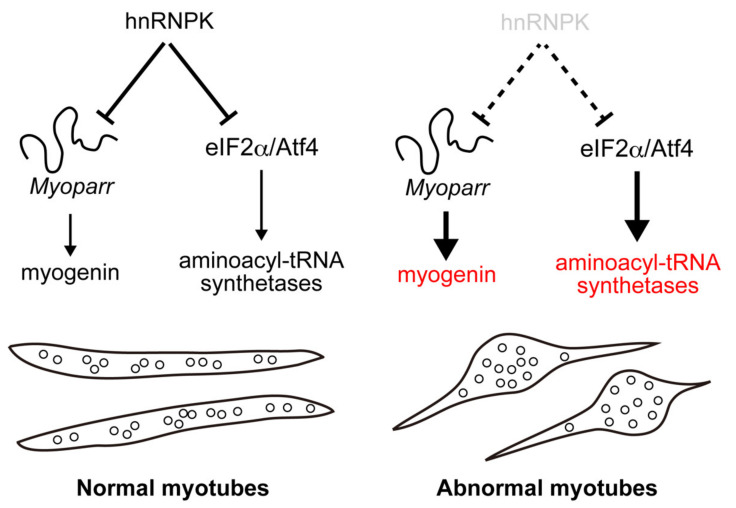
Two different downstream targets of hnRNPK during myogenic differentiation. During myogenic differentiation, hnRNPK has two different downstream targets. One is *myogenin*, which codes for a regulator of myogenic differentiation. hnRNPK represses the *myogenin* expression by binding to the ccawmcc motifs on *Myoparr*. The other target is aminoacyl-tRNA synthetases. hnRNPK regulates the expression of aminoacyl-tRNA synthetases via the eIF2α/Atf4 pathway. In hnRNPK-depleted cells, the hyperactivated expression of these genes may lead to the locally spherical formation of myotubes.

## Data Availability

The RNA-Seq raw data for each sample reported in this study were deposited into the DDBJ Sequence Read Archive under the accession no. DRA005527.

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
