# Peer review of "Myoparr-Associated and -Independent Multiple Roles of Heterogeneous Nuclear Ribonucleoprotein K during Skeletal Muscle Cell Differentiation"

_ijms, 2021, doi:10.3390/ijms23010108_

Round 1

Reviewer 1 Report

In this manuscript, the authors reported a series of the results describing the roles of heterogenous nuclear ribonucleoprotein K (hnRNPK) for muscle differentiation and its interactions with a long non-coding RNA Myoparr. The study was reasonably designed to test their hypothesis. The manuscript contains new data to explain the biological roles of hnRNPK and myoparr, which is valuable to the research community. While this reviewer does not have any major concerns in the current manuscript, there are several minor suggestions to bring better clarifications.

  1. Page 2, Line 82-83: This sentence sounds awkward. The authors intended to use EGFP mRNA as a negative control, so no binding of both molecules was already expected. This should be “…, EGFP mRNA could work as a negative control.” or any other revisions?

  1. Page 3, Section 2.2, the first paragraph: The phase “the regulatory role of hnRNPK in the expression of myogenin” seems to be overstated. The results in Figure 2A-C demonstrated the correlation of the expression for those two molecules but could not directly support the regulatory role of hnRNPK at this point.

  1. Page 4, Figure 2E: Please also clarify how many image fields were used to count all samples, not only describing the number of cells counted.

  1. Page 7, Section 2.4, the second paragraph and Page 8, Figure 4G: The analysis of myotube morphology was quite interesting. However, the images in Figure 4G are relatively small. It is great if enlarged images are added to show representative cellular morphology (locally spherical vs. tube-shaped), and/or if any indications (arrows and arrowheads etc.) are added in the current images to point out those myotubes.

  1. Page 12, Figure 7: The title of this figure sounds too definitive, while the present work remains preliminary.

  1. Page 9, Line 285: Please spell out the abbreviation ISRIB (integrated stress response inhibitor) when it first comes in the text.

  1. Pahe 14, Line 423: Please specify the catalogue number of ISRIB here. In the Cayman Chemical’s website, only trans-ISRIB is available. Was it the one which had been used for the studies?

Author Response

Author's Reply to the Review Report (Reviewer 1):

In this manuscript, the authors reported a series of the results describing the roles of heterogenous nuclear ribonucleoprotein K (hnRNPK) for muscle differentiation and its interactions with a long non-coding RNA Myoparr. The study was reasonably designed to test their hypothesis. The manuscript contains new data to explain the biological roles of hnRNPK and myoparr, which is valuable to the research community. While this reviewer does not have any major concerns in the current manuscript, there are several minor suggestions to bring better clarifications.

[Our reply]

We appreciate the reviewer's favourable comments and kind suggestions. According to the valuable comments and suggestions, we revised the original manuscript and showed corrections highlighted using the Track Changes function in Microsoft Word.

Comment 1

Page 2, Line 82-83: This sentence sounds awkward. The authors intended to use EGFP mRNA as a negative control, so no binding of both molecules was already expected. This should be “…, EGFP mRNA could work as a negative control.” or any other revisions?

[Our reply]

We appreciate your kind suggestion. We changed the sentence to "…, EGFP mRNA could work as a negative control." (Page 2, line 84).

Comment 2

Page 3, Section 2.2, the first paragraph: The phase “the regulatory role of hnRNPK in the expression of myogenin” seems to be overstated. The results in Figure 2A-C demonstrated the correlation of the expression for those two molecules but could not directly support the regulatory role of hnRNPK at this point.

[Our reply]

According to your suggestion, changed the sentence to ", demonstrating the correlation of the expression for those two genes." (Page 3, line 114).

Comment 3

Page 4, Figure 2E: Please also clarify how many image fields were used to count all samples, not only describing the number of cells counted.

[Our reply]

Thank you for your suggestion. We added the conditions (Page 4, line 147).

Comment 4

Page 7, Section 2.4, the second paragraph and Page 8, Figure 4G: The analysis of myotube morphology was quite interesting. However, the images in Figure 4G are relatively small. It is great if enlarged images are added to show representative cellular morphology (locally spherical vs. tube-shaped), and/or if any indications (arrows and arrowheads etc.) are added in the current images to point out those myotubes.

[Our reply]

We appreciate your comment "The analysis of myotube morphology was quite interesting". According to your valuable comments, we added enlarged images and arrowheads indicating locally spherical myotubes in Figure 4G (Page 8, lines 242-243).

Comment 5

Page 12, Figure 7: The title of this figure sounds too definitive, while the present work remains preliminary.

[Our reply]

According to your suggestion, we changed the sentence to "Two different downstream targets of hnRNPK during myogenic differentiation" (Page 12, line 341).

Comment 6

Page 9, Line 285: Please spell out the abbreviation ISRIB (integrated stress response inhibitor) when it first comes in the text.

[Our reply]

We added the abbreviation of ISRIB (Page 9, lines 283-284).

Comment 7

Pahe 14, Line 423: Please specify the catalogue number of ISRIB here. In the Cayman Chemical’s website, only trans-ISRIB is available. Was it the one which had been used for the studies?

[Our reply]

Yes, we used trans-ISRIB (No.16258, Cayman Chemical Company). We added the catalogue number of ISRIB (Page 14, line 427).

Reviewer 2 Report

In a previous study, the authors showed that the lncRNA Myoparr is involved in both the specification of myoblasts by activating the expression of myogenin (one of the key myogenic transcription factors) and in myoblast cell cycle withdrawal by triggering myogenic microRNA expression (Hitachi et al, Embo Rep 2019). Moreover, Myoparr promotes muscular atrophy caused by denervation and its knockdown rescues muscle wasting in mice. Finally, the authors identified Ddx17, hnRNPK and Tial1/TIAR as Myoparr-associated proteins. Here, they pursue by the characterization of the functions of hnRNPK in myogenic differentiation. The authors show that hnRNPK binds directly to Myoparr and identify the hnRNPK-binding region on Myoparr. They also observed that hnRNPK has a role in myogenic differentiation in a manner independent of Myoparr. The manuscript is well written and the experiments well conducted to address the questions on the role of hNRPK in myogenesis. However, no rescue experiments were conducted for the hnRNPK KD to exclude off-targets and as a mean of valuable control to regain the KD effects.

Line 29. It will be nice to clarify the sentence « …which do not encode more than 100 amino acids, are emerging… » Do the authors mean that ORFs are found? Indeed, a hundred amino acids correspond to a protein of about 10 kDa, which is not that small.

Line 160 and legend of figure 3. It will be nice to mention from which website the results of RBP-map analysis were generated.

Line 392. It will be nice to clarify the sentence. Why to mention those factors and not others?

Line 435. It is recommended by the MIQE guidelines to normalize the RT-qPCR using at least two genes. The authors used only Rpl26, why?

Finally, given the role of HDACs in muscle wasting and that hnRNPK can modulate HDAC expression in other cell system, it will be nice to include some information in the discussion. Were HDAC expression levels modulated in the hnRNPK KD RNA-seq?

Author Response

Author's Reply to the Review Report (Reviewer 2):

In a previous study, the authors showed that the lncRNA Myoparr is involved in both the specification of myoblasts by activating the expression of myogenin (one of the key myogenic transcription factors) and in myoblast cell cycle withdrawal by triggering myogenic microRNA expression (Hitachi et al, Embo Rep 2019). Moreover, Myoparr promotes muscular atrophy caused by denervation and its knockdown rescues muscle wasting in mice. Finally, the authors identified Ddx17, hnRNPK and Tial1/TIAR as Myoparr-associated proteins. Here, they pursue by the characterization of the functions of hnRNPK in myogenic differentiation. The authors show that hnRNPK binds directly to Myoparr and identify the hnRNPK-binding region on Myoparr. They also observed that hnRNPK has a role in myogenic differentiation in a manner independent of Myoparr. The manuscript is well written and the experiments well conducted to address the questions on the role of hNRPK in myogenesis. However, no rescue experiments were conducted for the hnRNPK KD to exclude off-targets and as a mean of valuable control to regain the KD effects.

[Our reply]

We appreciate the reviewer's comments and kind suggestions. According to your valuable comments and suggestions, we revised the original manuscript and showed corrections highlighted using the Track Changes function in Microsoft Word. To exclude the off-target effects by hnRNPK knockdown, we have used two independent siRNAs, which are more than 700 nt apart from each other, for hnRNPK knockdown. Off-target effects occur when the sense strand of siRNA having homology to untargeted genes was incorporated into RNA-induced silencing complex (RISC). Stealth siRNAs we used in this paper have modifications only to allow the antisense strand to efficiently enter the RISC, eliminating the concerns about sense-strand-derived off-target effects. In addition, Stealth siRNAs are chemically modified to minimize the induction of nonspecific cellular stress response pathways. The most advanced design algorithm using up-to-date bioinformatics also reduced off-target effects of Stealth siRNAs. Thus, off-target effects could be minimal in our experiments. I hope that the reviewer would kindly agree with our answer.

Comment 1

Line 29. It will be nice to clarify the sentence « …which do not encode more than 100 amino acids, are emerging… » Do the authors mean that ORFs are found? Indeed, a hundred amino acids correspond to a protein of about 10 kDa, which is not that small.

[Our reply]

We appreciate the reviewer's comments. We revised the sentence (Page 1, lines 28-29).

Comment 2

Line 160 and legend of figure 3. It will be nice to mention from which website the results of RBP-map analysis were generated.

[Our reply]

We added the URL of RBPmap to the Materials and Methods (Page 15, lines 513-514).

Comment 3

Line 392. It will be nice to clarify the sentence. Why to mention those factors and not others?

[Our reply]

We appreciate the reviewer's comments. We modified the sentences (Page 13, lines 391-393).

Comment 4

Line 435. It is recommended by the MIQE guidelines to normalize the RT-qPCR using at least two genes. The authors used only Rpl26, why?

[Our reply]

Regarding the normalization of RT-qPCR, we have referred to the similar analyses published in the IJMS (IJMS. 2018, 19, 2959; IJMS. 2019, 20, 3950; IJMS. 2021, 22, 503), in which the normalization of RT-qPCR was performed by one internal control. Quantitative gene expression data are normalized to the expression levels of Rpl26 in skeletal muscle (PLoS One. 2019, 14(4):e0215489). We have confirmed that Rpl26 expression was not significantly affected by hnRNPK knockdown, thus we concluded Rpl26 is good enough to use as an internal control.

Comment 5

Finally, given the role of HDACs in muscle wasting and that hnRNPK can modulate HDAC expression in other cell system, it will be nice to include some information in the discussion. Were HDAC expression levels modulated in the hnRNPK KD RNA-seq?

[Our reply]

We appreciate the reviewer's suggestion. According to your suggestion, we examined the expression changes of HDACs by hnRNPK knockdown. Our RNA-seq results showed that hnRNPK KD only decreased Hdac10 expression. Since the role of Hdac10 in muscle atrophy is unknown, we decided not to mention HDACs to prevent readers from misreading in this particular manuscript. The following list is the RNA-seq results of hnRNPK KD.

name       log2FoldChange      adjusted-pvalue

Hdac1     -0.061173541         0.912827

Hdac2     -0.232446042         0.50480063

Hdac3     -0.336635512         0.269896233

Hdac4     0.008657819          0.992838106

Hdac5     0.013421201          0.986968735

Hdac6     -0.134033454         0.798182568

Hdac7     -0.291840875         0.279863164

Hdac8     0.03814993            0.962220073

Hdac9     0.292866597          0.635584721

Hdac10    -0.944277271         8.97E-05

Hdac11    0.172861266          0.773441532